# Endogenous epitope-tagging of *Tet1*, *Tet2* and *Tet3* identifies TET2 as a naïve pluripotency marker

Raphaël Pantier , Tülin Tatar, Douglas Colby, Ian Chambers

*Tet1*, *Tet2*, and *Tet3* encode DNA demethylases that play critical roles during stem cell differentiation and reprogramming to pluripotency. Although all three genes are transcribed in pluripotent cells, little is known about the expression of the corresponding proteins. Here, we tagged all the endogenous *Tet* family alleles using CRISPR/Cas9, and characterised TET protein expression in distinct pluripotent cell culture conditions. Whereas TET1 is abundantly expressed in both naïve and primed pluripotent cells, TET2 expression is restricted to the naïve state. Moreover, TET2 is expressed heterogeneously in embryonic stem cells (ESCs) cultured in serum/leukemia inhibitory factor, with expression correlating with naïve pluripotency markers. FACS-sorting of ESCs carrying a *Tet2*$^{Flag-IRES-EGFP}$ reporter demonstrated that TET2-negative cells have lost the ability to form undifferentiated ESC colonies. We further show that TET2 binds to the transcription factor NANOG. We hypothesize that TET2 and NANOG co-localise on chromatin to regulate enhancers associated with naïve pluripotency genes.

## Introduction

Ten-eleven translocation (TET) family proteins are responsible for active DNA demethylation by sequential oxidation of 5-methylcytosine into 5-hydroxymethylcytosine, 5-formylcytosine, and 5-carboxylcytosine (1, 2). TET proteins contribute to DNA demethylation in naïve embryonic stem cells (ESCs) (3, 4, 5, 6) and their activity is required both for proper differentiation (7, 8) and for reprogramming to pluripotency (9). TET proteins are also critical for embryonic development, as *Tet1/2/3* triple-knockout embryos cannot proceed beyond gastrulation (10).

Although genetic studies indicate that TET proteins have redundant activities, the low level of sequence conservation outside the catalytic domain suggests that they may also exert distinct functions (11, 12). Indeed, Tet1, Tet2, and Tet3 have different expression patterns during development and in adult tissues (13). TET proteins also interact with partner proteins such as OGT and Sin3a complex members, which might promote functions independent of TET catalytic activity (14, 15, 16, 17).

Because of the lack of reliable commercial antibodies and reporter systems, TET protein expression, particularly at the single cell level, remains poorly characterized. In this study, we used CRISPR/Cas9 in ESCs to tag all endogenous *Tet1*, *Tet2*, and *Tet3* alleles with antibody epitopes and fluorescent reporters. These cellular reagents allowed the visualisation and the functional analysis of TET proteins in pluripotent cells.

## Results

### TET proteins present distinct expression patterns in ESCs

To visualise endogenous TET protein expression in ESCs, we generated knockin alleles using CRISPR/Cas9. Donor templates (targeting vectors or single-stranded oligonucleotides) were used to add epitope tags in frame with the TET protein coding sequences (Figs 1A and fig S1 for a summary of all cell lines). Initially, a targeting vector containing a puromycin resistance cassette (Puro$^R$) was used to add the triple Flag epitope tag (Flag)$_3$ at the C-terminus of TET1, resulting in the generation of heterozygous *Tet1*$^{Flag-IP/+}$ ESC clones (Fig S2). To obtain a cell line expressing only tagged versions of TET1, the remaining wild-type allele of *Tet1*$^{Flag-IP/+}$ clone C10 was re-targeted using a vector with an EGFP reporter to give *Tet1*$^{Flag-IP/Flag-IGFP}$ cells (Fig S3). For subsequent modifications of *Tet* alleles, single-stranded DNA (ssDNA) oligonucleotides were used as donor templates for homologous recombination, as they result in high targeting efficiencies and do not require the use of a selection cassette (18). This alternative strategy was used to fuse a V5 epitope to the C terminus of TET2 in *Tet1*$^{Flag-IP/Flag-IGFP}$ clone C1 (Fig S4). In clones C2 and C3, both *Tet2* alleles were successfully modified in a single step, resulting in the generation of double-tagged *Tet1*$^{Flag-IP/Flag-IGFP}$; *Tet2*$^{V5/V5}$ ESC clones. To generate an ESC line carrying all six modified *Tet* alleles, *Tet1*$^{Flag-IP/Flag-IGFP}$; *Tet2*$^{V5/V5}$ clone C3 was modified using a ssDNA that fused a HA tag to the C terminus of TET3. PCR genotyping identified two *Tet1*$^{Flag-IP/Flag-IGFP}$; *Tet2*$^{V5/V5}$; *Tet3*$^{HA/HA}$ clones (Fig S5), and

UK Medical Research Council Centre for Regenerative Medicine, Institute for Stem Cell Research, School of Biological Sciences, University of Edinburgh, Edinburgh, Scotland

Correspondence: ichambers@ed.ac.uk

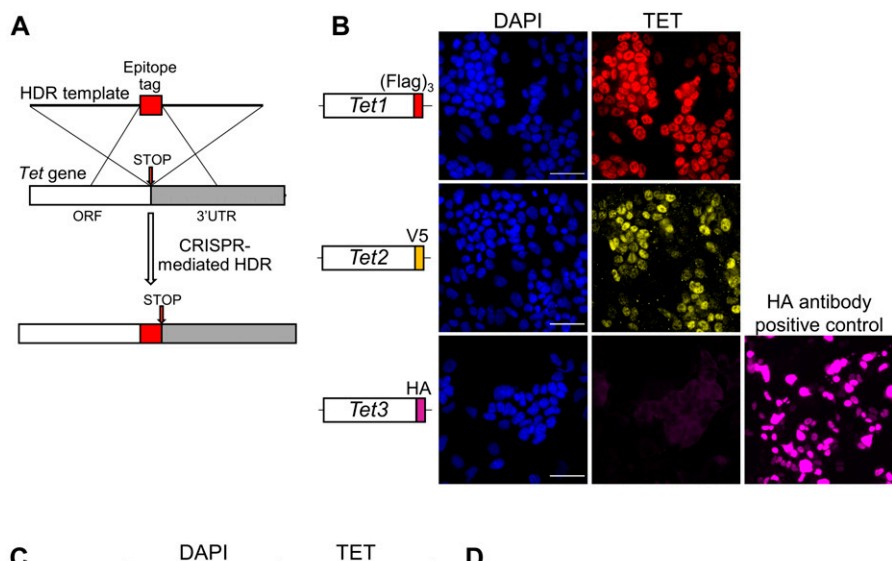

Figure 1.   TET protein expression and heterogeneity in ESCs.
**(A)** General strategy for generating tagged *Tet* knockin alleles. ESCs were co-transfected with a gRNA designed near the stop codon and a repair template (single-stranded oligo or targeting vector) containing an epitope tag (Flag, V5 or HA). **(B, C)** Immunofluorescence for Flag (TET1, red), V5 (TET2, yellow), and HA (TET3, magenta) in *Tet*^tag/tag^ ESCs cultured in serum/LIF (B) or 2i/LIF (C). **(B)** Wild-type E14Tg2a ESCs transfected with an HA-NANOG expression plasmid provided a positive control (B). Scale bars: 50 *μm*. **(D)** Immunofluorescence for V5 (yellow) in *Tet1*^V5/V5^ ESCs (top) and *Tet*^tag/tag^ ESCs (bottom) cultured in serum/ LIF. Samples were imaged and processed under the same conditions to allow a direct comparison of TET1 and TET2 expression levels. Scale bars: 50 *μm*.

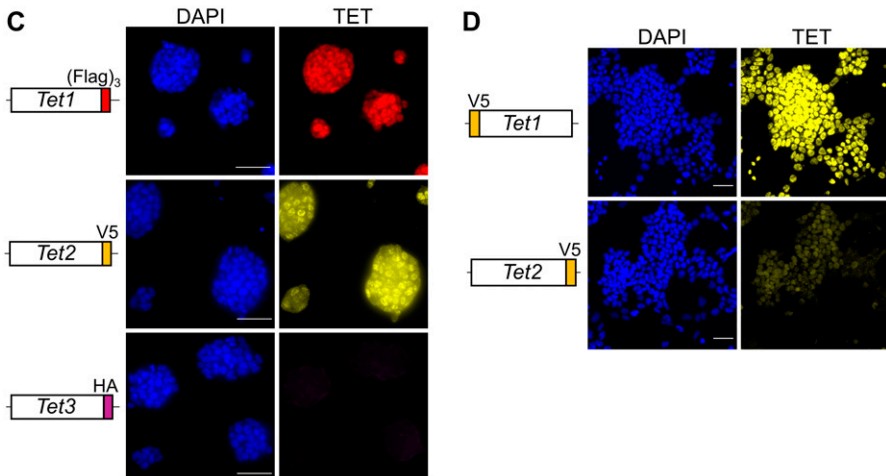

Sanger sequencing confirmed that clone *C7* has both *Tet3* alleles appropriately modified, which we refer to as *Tet*^tag/tag^ ESCs.

To investigate the expression profile and single cell heterogeneity of TET1, TET2, and TET3 proteins, we performed immunofluorescence analyses on *Tet*^tag/tag^ ESCs using antibodies recognising flag (TET1), V5 (TET2), or HA (TET3) epitope tags. In serum/Leukemia inhibitory factor (LIF), TET1 is expressed in most cells at relatively homogenous levels within the population (Fig 1B). In contrast, TET2 is heterogeneously expressed with a "salt and pepper" pattern composed of a mixture of TET2-positive and TET2-negative ESCs (Fig 1B). TET3 was undetectable (Fig 1B). To further explore TET protein expression in the naïve state, we cultured *Tet*^tag/tag^ ESCs in the presence of LIF and inhibitors of MEK and GSK3β (2i/LIF). In this condition, ESCs form dome-shaped colonies and homogenously express naïve pluripotency markers (19). In 2i/LIF, both TET1 and TET2 present a more homogenous expression pattern (Fig 1C). Once again TET3 protein was undetectable (Fig 1C). These data on TET protein expression are in accord with the relative expression of *Tet* mRNAs (Fig S6A and B). Interestingly, all *Tet* mRNAs are expressed at lower levels in 2i/LIF than serum/LIF.

Although the preceding analyses demonstrated that TET family proteins are differentially expressed in ESCs, the use of different epitope tags did not allow a direct comparison of expression levels of different TET proteins. Therefore, to allow the relative quantification of TET1 and TET2 proteins, we targeted *Tet1* in E14Tg2a ESCs with an ssDNA to introduce the V5 epitope tag (Fig S7A). Two clones in which both *Tet1* alleles were tagged by V5 were obtained (Fig S7B). We next performed comparative analyses of *Tet1*^V5/V5^ ESCs with *Tet*^tag/tag^ ESCs in which an identical V5 epitope tag was fused to TET2. We confirmed that both cell lines retain similar self-renewal efficiencies as wild-type E14Tg2a ESCs (Fig S8A) and express normal levels of the pluripotency factor NANOG (Fig S8B and C). Comparative immunostaining of *Tet1*^V5/V5^ ESCs with *Tet*^tag/tag^ ESCs using a V5 antibody showed that TET1 is expressed at much higher levels than TET2 in serum/LIF (Fig 1D). Consistent with low protein abundance, TET2 was undetectable by Western blot but could be detected after enrichment by immunoprecipitation, showing a band at the predicted size (210 kD) for the full-length TET2-V5 protein (Fig S6C).

Together, these analyses revealed for the first time the relative expression of TET proteins expression at the single cell level in ESCs.

## TET2 marks self-renewing ESCs in serum/LIF

To further characterise TET2 function, we generated a $Tet2^{Flag-IRES-EGFP}$ reporter cell line from E14Tg2a ESCs. After transfection with CRISPR/Cas9 and a targeting vector, EGFP⁺ ESCs were sorted into single wells and expanded (Fig S9A). PCR analyses identified nine ESC clones in which (Flag)₃-IRES-EGFP was targeted to the 3′ end of $Tet2$ (Fig S9B). Of these, four clones did not produce PCR products of wild-type alleles, indicating that both $Tet2$ alleles were modified (Fig S9C). Co-immunofluorescence analysis of ESCs cultured in serum/LIF confirmed the heterogeneous TET2 expression pattern and indicated that TET2 (Flag) expressing cells were also fluorescently marked by the cytoplasmic EGFP transcriptional reporter (Fig 2A). FACS of ESCs cultured in serum/LIF allowed the selection of $Tet2^{Flag-IRES-EGFP}$ cells based on their TET2 expression level, using the nonfluorescent parental cell line (E14Tg2a) as a negative control (Figs 2B and Fig S10). To investigate the self-renewal efficiency of ESCs expressing distinct TET2 levels, ESCs cultured in serum/LIF were FACS-sorted into TET2-positive and TET2-negative populations. The cells were then plated at clonal density and stained for AP following 7 d of culture in serum/LIF condition. Strikingly, TET2-negative ESCs formed almost no self-renewing ESC colonies (Fig 2C). In contrast, TET2-positive ESCs showed a similar number and proportion of AP-stained colonies compared with the bulk-sorted control and parental cell lines (Fig 2C).

To examine transcriptional differences between TET2-positive and TET2-negative ESC populations, quantitative reverse transcription PCR (RT-qPCR) analysis was performed on selected transcripts (Fig 2D). As expected, $Tet2$ mRNA expression was high in TET2-positive ESCs, and dramatically decreased in TET2-negative ESCs. The transcript levels of the pluripotency factor $Oct4$ were decreased by 50% in TET2-negative ESCs compared with the bulk-sorted control or the parental cell line. This reduction was more pronounced with the naïve markers $Nanog$ and $Esrrb$.

Together, these data suggest that TET2 expression is tightly associated with naïve pluripotency marker expression and efficient ESC self-renewal.

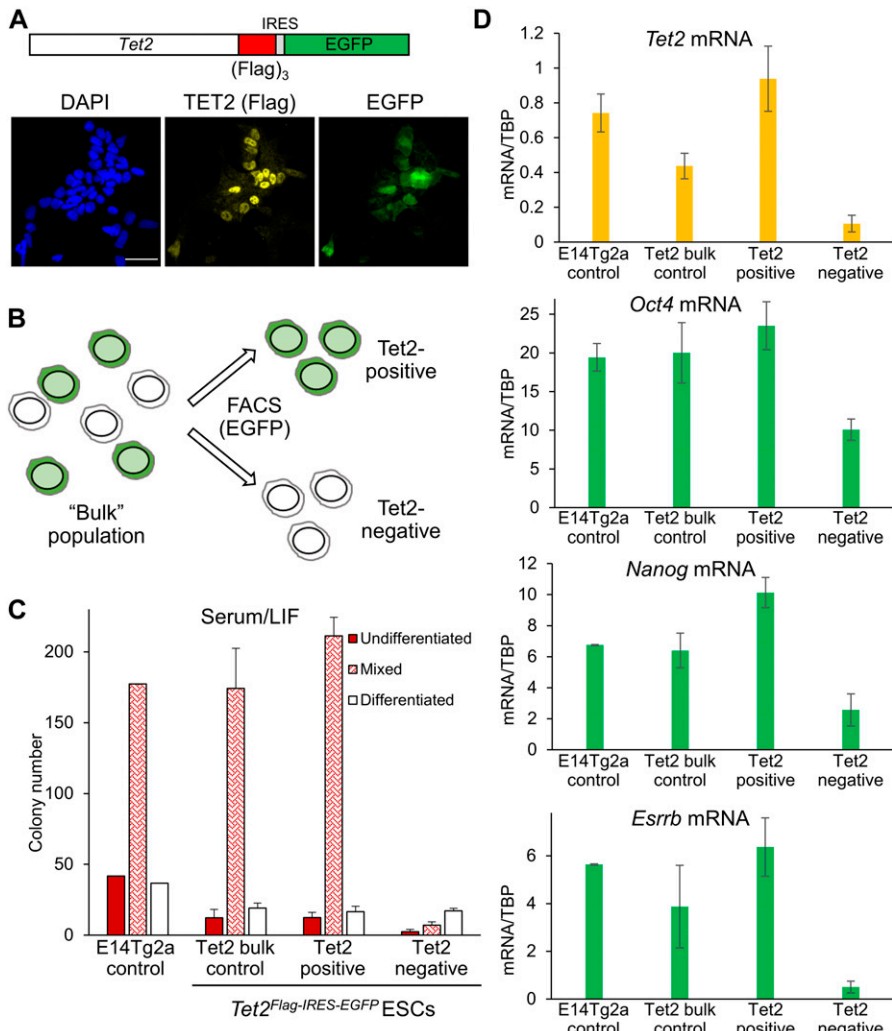

**Figure 2. TET2 marks self-renewing ESCs in serum/LIF culture condition.**
**(A)** Co-immunofluorescence for Flag (TET2, yellow) and EGFP (green) in $Tet2^{Flag-IRES-EGFP}$ ESCs cultured in serum/LIF. Scale bar: 50 μm. **(B)** General strategy for sorting TET2-EGFP–positive and TET2-EGFP–negative $Tet2^{Flag-IRES-EGFP}$ ESCs. **(C)** Clonal self-renewal assays of FACS-sorted $Tet2^{Flag-IRES-EGFP}$ ESCs (or the wild-type parental cell line). Error bars: SD of the mean (n = 3). **(D)** Quantitative mRNA expression in FACS-sorted $Tet2^{Flag-IRES-EGFP}$ ESCs, compared with the wild-type parental cell line. Error bars: SD of the mean (n = 2).

### TET2 interacts with NANOG and co-localises at ESC enhancers

NANOG is heterogeneously expressed in ESCs and its expression level is directly related to self-renewal efficiency (20, 21). As both TET2 and NANOG behave as naïve pluripotency markers and were reported to interact with each other (22), we further investigated the relationship between these two proteins.

To compare the expression patterns of TET2 and NANOG, we performed co-immunofluorescence in ESCs grown in serum/LIF (Fig 3A). Quantitation of immunofluorescence in single cells showed that the vast majority of TET2-positive cells co-express NANOG (Fig 3B). This observation further confirms the correlation between TET2 and NANOG that we identified at the mRNA expression level (Fig 2D). However, TET2 marks a larger population of cells than NANOG, resulting in the detection of TET2⁺/NANOG⁻ ESCs (Fig 3A and B).

To examine the physical interaction between TET2 and NANOG, we performed co-immunoprecipitations in ESCs using differently truncated TET2 constructs (Fig S11A). Interestingly, both TET2 N terminus (1–1,221) and C terminus (924–1,911) interact with NANOG (Fig S11B). Non-overlapping TET2 truncations (1–828 and 924–1,377)

retained their interaction with NANOG (Fig S11C). This suggests the presence of at least two NANOG-binding regions within TET2. Interestingly, the stronger interaction with full-length TET2 compared with TET2 N- and C-terminal fragments (Fig S11C), suggests that NANOG-binding regions act in a cooperative manner.

To further explore the interaction between TET2 and NANOG, we investigated the chromatin-binding profile of these two proteins in ESCs. Previously published TET2 (23) and NANOG (24) ChIP-seq datasets were analysed and compared, with a particular focus on the pluripotency gene regulatory network which controls ESC self-renewal (25). TET2 and NANOG ChIP-seq signals (and their respective input controls) were visualised as heat maps at ESC enhancers, defined as sites co-bound by OCT4-SOX2-NANOG (24). Both TET2 and NANOG are centrally enriched at ESC enhancers (Fig 3C). To validate this observation, we examined TET2 and NANOG ChIP-seq signal at relevant pluripotency genes (*Nanog*, *Oct4*, *Esrrb*, *Klf4*, and *Prdm14*) using a genome browser. Interestingly, TET2- and NANOG-binding profiles are highly similar at these loci with most peaks co-localising within ESC "super-enhancers" (24) (Fig 3D).

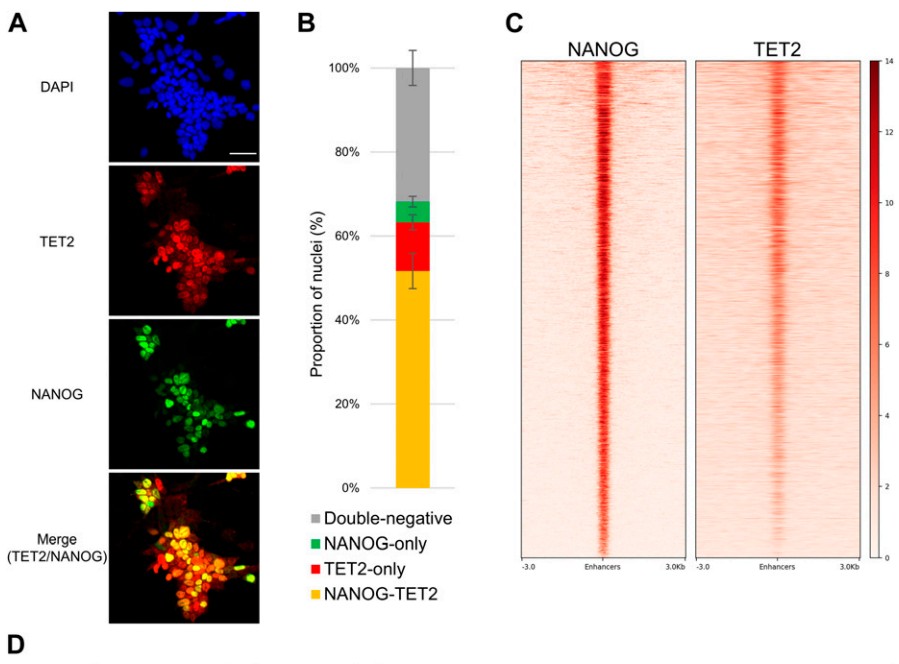

**Figure 3.** **TET2 interacts with NANOG and co-localises at ESC enhancers.**
**(A)** Co-immunofluorescence for Flag (red) and NANOG (green) in *Tet2^Flag-IRES-EGFP* ESCs cultured in serum/LIF. Scale bar: 50 μm. **(B)** Quantification of TET2/NANOG co-immunofluorescence (as in panel [A]) in four independent *Tet2^Flag-IRES-EGFP* ESC clones cultured in serum/LIF. For each clone, nuclei were counted in two independent fields of view. **(C)** Heat maps showing NANOG (24) and TET2 (23) RPKM-normalised ChIP-seq signal at ESC enhancers. **(D)** Genomic snapshots showing TET2 (red) and NANOG (green) ChIP-seq signal at selected pluripotency loci. Blue bars: ESC super-enhancers (24).

Together, these results indicate that TET2 and NANOG physically interact and co-localise on chromatin to regulate the pluripotency gene regulatory network.

### TET proteins are dynamically expressed during the transition from naïve to primed pluripotency

Recent studies demonstrated that ESCs can be driven from a naïve to a primed pluripotent state in vitro, reflecting a change from a pre- to post-implantation epiblast molecular signature. This transition is accompanied by global epigenomic and transcriptional changes (26), to which TET proteins might contribute (27).

To examine the expression of *Tet* family genes in the primed pluripotent state, we used two different culture systems: Epiblast-like cells (EpiLC) (28) and Epiblast stem cells (EpiSC) (29). First, we examined the mRNA levels of *Tet1*, *Tet2*, and *Tet3* by RT-qPCR (Fig 4A). In EpiLCs (24 and 48 h), *Tet1* transcript levels are similar to naïve ESCs cultured in 2i/LIF. In contrast, *Tet1* mRNA is decreased by threefold to fivefold in EpiSCs compared with ESCs cultured in serum/LIF and 2i/LIF, respectively. *Tet2* transcripts are dramatically decreased both in EpiLCs

and EpiSCs compared with ESCs, further confirming that *Tet2* behaves as a naïve pluripotency marker. In EpiLCs (48 h) and EpiSCs, *Tet3* is transcribed at levels which are similar to ESCs cultured in serum/LIF, where TET3 protein is undetectable (Fig 1B).

To extend our analysis to the expression of TET proteins in the primed state, we performed EpiLC and EpiSC differentiation with our *Tet^{tag/tag}* ESC line, followed by immunofluorescence for TET1 (Flag), TET2 (V5), or TET3 (HA), together with the control marker OCT6 (26). Both in EpiLCs (Fig 4B) and in EpiSCs (Fig 4C), TET1 was the only detected protein, showing a homogenous expression pattern. However, a long exposure time was required to image TET1 in EpiSCs, indicating a lower expression level than EpiLCs/ESCs and confirming the results from RT-qPCR analysis (Fig 4A).

To further explore changes in TET protein expression during the transition from naïve to primed pluripotency, we performed a time course EpiSC differentiation experiment with *Tet^{tag/tag}* ESCs. Surprisingly, TET1 protein showed highly dynamic changes (Fig 4D). Up to day 3 of the EpiSC differentiation protocol, TET1 expression was progressively lost in the whole population. On day 4, both TET1 and NANOG were re-expressed at high levels in a subset of cells, which

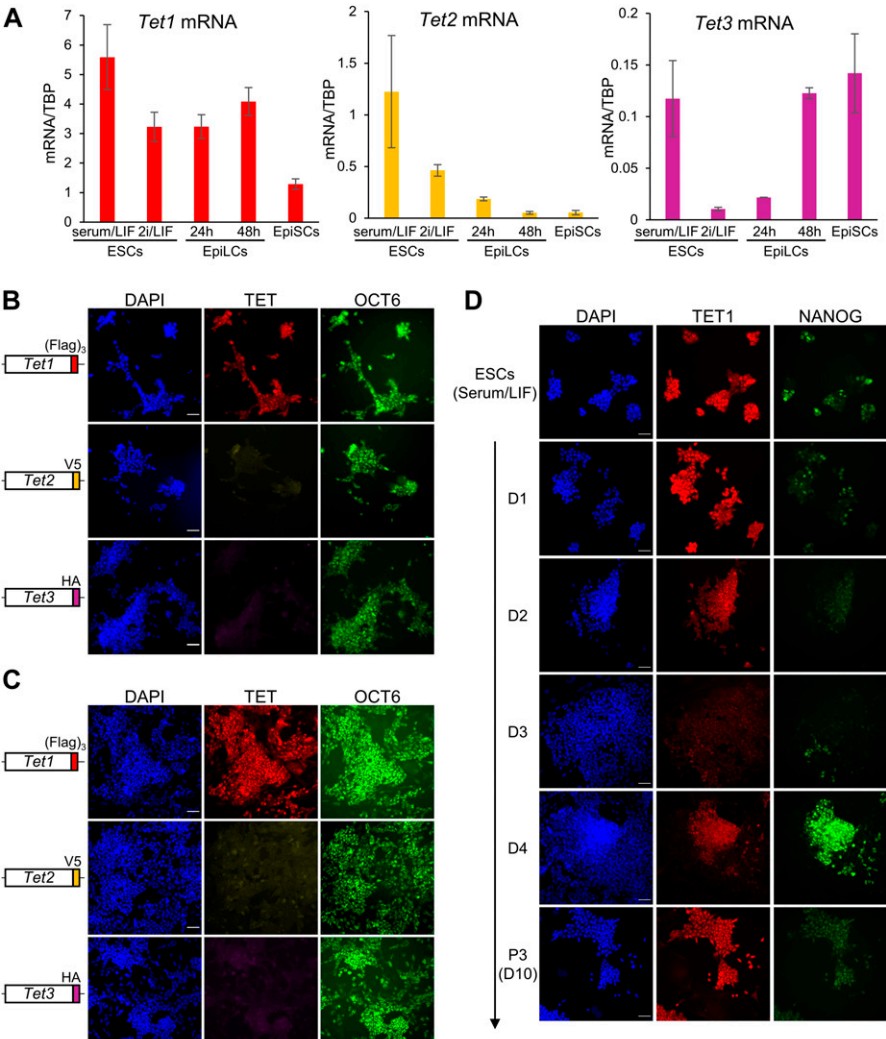

**Figure 4. TET protein dynamics during the transition from naïve to primed pluripotency.**
**(A)** Quantitative mRNA expression in E14Tg2a ESCs cultured in naïve conditions (serum/LIF or 2i/LIF) and differentiated into primed EpiLCs (24 and 48 h) or EpiSCs (passage 15). Error bars: SD of the mean (n ≥ 2). **(B, C)** Co-immunofluorescence for OCT6 (green) and Flag (TET1, red) or V5 (TET2, yellow) or HA (TET3, magenta) in *Tet^{tag/tag}* EpiLCs (48 h) (B) or in *Tet^{tag/tag}* EpiSCs (passage 11) (C) cultured in activin/FGF. Scale bars: 50 μm. **(D)** Time course co-immunofluorescence for Flag (TET1, red) and NANOG (green) in *Tet^{tag/tag}* cells during EpiSC differentiation (arrow) from serum/LIF culture condition. Samples were imaged and processed under the same conditions to allow a direct comparison of TET1 and NANOG expression levels between different time points. Scale bars: 50 μm.

may mark future EpiSCs, as TET1 and NANOG expression was homogenous by passage 3 (at lower levels than day 4). Interestingly, TET2 followed a similar expression dynamics during early EpiSC differentiation (Fig S12A), although its expression was completely lost in stable EpiSCs (Fig 4C). In contrast, TET3 remained undetectable throughout EpiSC differentiation (Fig S12B). A similar time course immunofluorescence experiment was performed in $Tet^{tag/tag}$ cells during EpiLC differentiation (from 24 to 72 h). This analysis did not reveal any change in TET1 protein expression compared with 2i/LIF (Fig S13A), whereas TET2 and TET3 could not be detected at any time point (Fig S13B and C), in agreement with their low transcription levels (Fig 4A).

Together, these experiments revealed dynamic changes in TET protein expression during the transition from naïve to primed pluripotency.

# Discussion

In this study, we comparatively assessed the expression of all *Tet* family genes in pluripotent cells. Immunofluorescence analyses revealed that TET1 protein is expressed in both naïve and primed pluripotent cells, whereas TET2 is exclusively expressed in naïve ESCs. Furthermore, TET1 is homogeneously expressed in serum/LIF, whereas TET2 is expressed only in a subset of cells. This differential expression suggests that TET1 and TET2 might exert distinct functions in pluripotent cells, which is supported by recent studies. Knockdown experiments indicated that TET1 and TET2 control the methylation of distinct genomic regions in ESCs (11). In addition, the ectopic expression of TET2 (but not TET1) reprograms cells from primed to naïve pluripotency (12).

Our work indicated that TET2 is a naïve pluripotency marker. In serum/LIF, TET2 specifically marks self-renewing ESCs and is associated with the expression of other heterogeneously expressed factors controlling the naïve state like NANOG (21) and Esrrb (30). Conversely, TET2-negative cells present low levels of naïve pluripotency markers and are unable to form AP-positive colonies. In addition, TET2 expression becomes homogenous during the transition from serum/LIF to 2i/LIF, which maintains ESCs in a naïve state. Conversely, TET2 expression is abolished during the transition to the primed state, either with the EpiLC or the EpiSC differentiation protocol. Further molecular characterisation on TET2-positive and TET2-negative ESCs will be required to reveal genome-wide transcriptional and methylation changes between these two cell populations.

We demonstrated that TET2 and NANOG are mostly co-expressed in ESCs cultured in serum/LIF. This observation could be extended to the analysis of TET2 expression in the inner cell mass of the pre-implantation blastocyst, where NANOG presents a "salt and pepper" expression pattern, which is critical to determine the balance between epiblast and primitive endoderm (31). Furthermore, NANOG transiently fluctuates between high and low expression states in ESCs (21). As TET2 is a direct transcriptional target of NANOG (32), its expression might fluctuate in a similar manner, explaining the correlation between TET2 and NANOG expression in ESCs.

In this study, we have shown that TET2 physically interacts with NANOG and co-localises at pluripotency enhancers. As TET2 lacks a DNA-binding domain, it might be targeted by NANOG to these loci to maintain them in a demethylated state. A recent study reported that TET2 could be efficiently ChIPed only after extensive cross-linking with formaldehyde plus disuccinimidyl glutarate (33), suggesting that TET2 interacts indirectly with chromatin. TET2 localises to completely different loci in ESCs and hematopoietic cells (33), supporting a model in which TET2 is targeted to chromatin via protein–protein interactions in a context-dependent manner. In contrast, TET1 has a CXXC domain, which recruits this protein mostly to CpG islands (17, 34). Supporting this model, *Tet2* knockout causes hypermethylation of enhancers in ESCs, whereas *Tet1* knockout has no effect on the methylation status of these genomic regions (35). More recently, TET2 was shown to promote enhancer demethylation by interacting with C/EBPα, Klf4 and Tfcp2l1 at distinct stages of induced pluripotent stem cell reprogramming (36).

Here, time course experiments allowed the visualisation of TET expression changes during the transition from naïve to primed pluripotency. Interestingly, TET1/2 and NANOG showed similar dynamics during early EpiSC differentiation. The global and transient decrease in NANOG/TET expression might mark the recently described "formative" pluripotent state (37). However, TET proteins did not show dynamic expression changes during EpiLC differentiation, which might be explained by a rapid and homogenous transition to the primed state (28) without passing through a "formative" pluripotent state. In addition, TET1 is expressed at higher levels in EpiLCs compared with EpiSCs. These observations highlight differences between the EpiLC and EpiSC differentiation protocols.

In conclusion, we generated knockin alleles of *Tet* family genes with epitope tags or fluorescent reporters, providing a robust characterisation of TET protein expression dynamics and single-cell heterogeneity in pluripotent cells. The engineered ESC lines produced in this study could be further exploited to study TET1/2/3 protein expression in vivo in transgenic mouse models.

# Materials and Methods

### Cell culture

All the cell lines in this study were derived from E14Tg2a (38) and incubated in a 37°C/7% $CO_2$ incubator. ESCs were cultured on gelatin-coated plates. Composition of the serum/LIF medium: Glasgow Minimum Essential Medium (Cat. no. G5154; Sigma-Aldrich), 10% fetal bovine serum, 1× L-glutamine (Cat. no. 25030-024; Invitrogen), 1× pyruvate solution (Cat. no. 11360-039; Invitrogen), 1× MEM nonessential amino acids (Cat. no. 11140-036; Invitrogen), 0.1 mM 2-mercaptoethanol (ref. 31350010; Gibco), and 100 U/ml LIF (made in-house).

For 2i/LIF ESC culture (19), serum-free N2B27 medium was prepared: 1:1 vol/vol mix of DMEM:F12 (Cat. no. 12634010; Gibco) and Neurobasal (Cat. no. 21103049; Gibco), 1× L-glutamine (Cat. no. 25030-024; Invitrogen), 1× MEM nonessential amino acids (Cat. no. 11140-036; Invitrogen), 0.1 mM 2-mercaptoethanol (ref. 31350010; Gibco), 1× N2 supplement (Cat. no. 17502048; Gibco), 1× B27 supplement (Cat. no. 17504044; Gibco). 1 μM PD0325901 (Cat. no. 1408; Axon), 3 μM CHIR99021 (Cat. no. 1386; Axon), and 100 U/ml LIF were added freshly to the medium.

EpiSC lines were derived in vitro from ESCs (29). 3 × 10⁴ ESCs were plated in a well of a six-wells plate with serum/LIF medium (see composition above). After 24 h, the medium was switched to N2B27 medium (see composition above) supplemented with 20 ng/ml human activin A (Cat. no. 120-14E; PeproTech) and 10 ng/ml human Fgf basic (Cat. no. 233-FB-025/CF; R&D Systems). The cells were submitted to daily media changes and passaged at day 5 of the protocol in six-well plates coated with 7.5 µg/ml bovine fibronectin. The cells were maintained in N2B27 medium supplemented with Activin/Fgf and passaged every 2–3 d. Homogenous EpiSCs were derived within 10 passages.

EpiLC differentiation was performed as described in (39). ESCs were adapted to 2i/LIF culture for at least three passages on poly-L-ornithine (Cat. no. P3655; Sigma-Aldrich) and laminin-coated wells (Cat. no. 354232; BD Biosciences) of a six-well plate. 2.5 × 10⁵ ESCs were plated on a well of six-well plate pretreated with 16.6 µl/ml fibronectin (Cat. no. FC010; Millipore) and containing EpiLC medium: N2B27 medium (see the composition above) supplemented with 20 ng/ml human activin A (Cat. no. 120-14E; PeproTech), 12 ng/ml human Fgf basic (Cat. no. 233-FB-025/CF; R&D Systems) and 1% knockout serum replacement (Cat. no. 10828-028; Gibco). The cells were submitted to daily media changes and collected for analyses.

### Self-renewal assays

Cells were collected by trypsinisation and resuspended in PBS (Cat. no. D8537; Sigma-Aldrich) supplemented with 2% fetal bovine serum at a concentration of around 1 × 10⁶ cells/ml. The cell suspension was passed through a cell strainer and kept on ice until cell sorting (-FACSAria II; Becton Dickinson). Single cells were gated using the forward-scattered light and side-scattered light parameters. Auto-fluorescent (dead) cells were also discarded. Gates for selecting EGFP fluorescent cells ("GFP B 525/50") were drawn using the non-fluorescent parental cell line (E14Tg2a ESCs) as a negative control. Tet2 bulk and E14Tg2a WT control cells were processed by FACS with no selection based on EGFP fluorescence. 600 cells were directly sorted in gelatin-coated wells of six-well plates containing serum/LIF medium (see the composition above). After 7 d of culture, the cells were washed in PBS and incubated for 1 min in a fixative solution made by mixing 25 ml of citrate solution (18 mM citric acid, 9 mM sodium citrate, and 12 mM NaCl), 8 ml of formaldehyde solution (37% vol/vol in water), and 65 ml of acetone. Fixed cells were washed in distilled water and stained for AP expression using a leukocyte AP kit (Cat. no. 86R-1KT; Sigma-Aldrich). Colonies were counted and categorised according to their morphology and AP staining.

### CRISPR-mediated homologous recombination

To modify endogenous Tet1/2/3 alleles, a double-strand break was generated at desired genomic loci using Cas9 and a synthetic gRNA (18). gRNAs were designed (http://crispr.mit.edu/) and cloned into Cas9/gRNA co-expression plasmids (pX330; Addgene, or derivatives).

To prevent cutting of targeted alleles by CRISPR/Cas9, donor templates were designed so that the gRNA site is disrupted after homologous recombination. Alternatively, a silent mutation was added in the donor template to disrupt the gRNA PAM sequence

(NGG). Two types of donor templates were used for homologous recombination: targeting vector or ssDNA oligonucleotide. Targeting vectors were cloned by Gibson assembly into a pBluescript backbone and contained a selection cassette (fluorescent reporter or puromycin resistance). The 5'- and -3' homology arms (typical size around 1.5 Kb) were amplified by PCR from ESC genomic DNA. ssDNAs (presenting around 60-bp homology arms) were ordered from Integrated DNA Technologies as Ultramer DNA oligonucleotides.

1 × 10⁶ ESCs were transfected using Lipofectamine 3000 (cat. L3000008; Thermo Fisher Scientific), following the manufacturer's instructions, with both the Cas9/gRNA plasmid and the donor template (targeting vector or ssDNA). After 48 h, ESCs were selected either by FACS sorting (targeting vector with fluorescent reporter or fluorescent Cas9) or by the addition of 0.75 µg/ml puromycin (targeting vector with puromycin resistance cassette). ESC clones were expanded in 24-well plates and genomic DNA was extracted (Cat. no. 69506; QIAGEN) for genotyping Tet alleles. PCR genotyping was performed using forward and/or reverse primers binding outside the homology arms of the donor template, therefore confirming the modification of the endogenous locus. PCR products were submitted to Sanger sequencing to confirm that the desired modification was added in frame with the Tet coding sequence. Correctly targeted ESC clones were expanded, and frozen aliquots were transferred to liquid nitrogen tanks for long term storage.

### List of gRNAs used for tagging Tet alleles

| Target locus | gRNA site (not including PAM sequence) |
|---|---|
| Tet1 start codon | TTTGGAAGGCTTTGCGGGGC |
| Tet1 stop codon | TGCGGGACCCTACAATCGTT |
| Tet2 stop codon | ACAACACATTTGTATGACGC |
| Tet3 stop codon | AGCCGCTGGATCTAGGTGCC |

### List of genotyping primers

| Target locus | Primer sequence |
|---|---|
| Tet1 3' FW1 | CTGATGTATCCCCCGAAGCC |
| Tet1 3' FW2 | CCACGTCCTGCCACTATACC |
| Tet1 3' RV1 | TCGGAGTTGAAATGGGCGAA |
| Tet1 3' RV2 | GGGCTTCTTGTGGCATCTCT |
| Puro FW | GCCGCGCAGCAACAGATGGAA |
| Puro RV | ACCCACACCTTGCCGATGTC |
| EGFP RV | AACTTCAGGGTCAGCTTGCC |
| Tet2 3' FW | ACAGGGTCTGTGACTACGGA |
| Tet2 3' RV1 | ACAGATGCTGTGACCTGTCC |
| Tet2 3' RV2 | CTGTGTCCCACGGTTACACA |
| Tet3 3' FW | CCGTGTCCTCTTACGCCTAC |
| Tet3 3' RV | CATGAGGGCAAAAGCACCAC |
| Tet1 5' FW | ACTCCGATGATCCTGCCTCT |
| Tet1 5' RV | TCGGGGTTTTGTCTTCCGTT |

### Immunofluorescence analysis

Cells were washed with PBS and fixed with 4% PFA for 10 min at room temperature. After fixation, the cells were washed with PBS and permeabilised with a solution of PBS containing 0.3% (vol/vol) Triton X-100 for 10 min at room temperature. Samples were blocked in blocking buffer (PBS supplemented with 0.1% [vol/vol] Triton X-100, 1% [wt/vol] BSA, and 3% [vol/vol] serum of the same species as the secondary antibodies were raised in) for 1 h at room temperature. After blocking, the samples were incubated with primary antibodies diluted in blocking buffer overnight at 4°C. After four washes with PBS containing 0.1% (vol/vol) Triton X-100, the samples were incubated with fluorescently labelled secondary antibodies diluted in blocking buffer for 1 h at room temperature in the dark. The cells were washed four times with PBS containing 0.1% (vol/vol) Triton X-100. DNA was stained with DAPI for 5 min at room temperature. The cells were washed with PBS for 5 min. The samples were imaged by fluorescence microscopy (Ti-E; Nikon). Images were analysed and processed using the software Fiji.

### Immunoprecipitation

E14/T ESCs were used for producing proteins for immunoprecipitation, as they can replicate and propagate pPyCAG plasmids which carry a polyoma origin of replication ([20]). $3 \times 10^6$ E14/T ESCs were transfected using Lipofectamine 3000 (Cat. no. L3000008; Thermo Fisher Scientific), following the manufacturer's instructions, with 6 μg of pPYCAG plasmids carrying a construct of interest.

E14/T ESCs were harvested 24 h after transfection. The cells were trypsinised, pelleted (5 min, 393$g$, 4°C), and washed twice with cold PBS before lysis in a swelling buffer (5 mM Pipes, pH 8, and 85 mM KCl) freshly supplemented with 1× protease inhibitor cocktail (Cat. no. 04 693 116 001; Roche) and 0.5% NP-40. After 20 min on ice with occasional shaking, nuclei were pelleted (10 min, 524$g$, 4°C) and resuspended in 1 ml of lysis buffer (20 mM Hepes, pH 7.6, 350 mM KCl, 0.2 mM EDTA, pH 8, 1.5 mM MgCl$_2$, and 20% glycerol) freshly supplemented with 0.2% NP-40, 0.5 mM DTT, and 1X protease inhibitor cocktail (Cat. no. 04 693 116 001; Roche). The material was transferred into nonstick microtubes (Cat. no. LW2410AS; Alpha Laboratories) and supplemented with 150 U/ml

of Benzonase Nuclease (Cat. no. 71206; Novagen). The samples were incubated on a rotating wheel for 30 min at 4°C. Tubes were centrifuged (16,100$g$, 30 min, 4°C) and nuclear extracts were collected in clean nonstick tubes. 30–50 μl of nuclear protein extract was boiled in Laemmli buffer as input material.

For immunoprecipitations, 5 μg of V5 antibody (Cat. no. 14-6796-80; eBioscience) or Flag antibody (Cat. no. F3165; Sigma-Aldrich) was added to nuclear extracts. For negative controls, 5 μg of normal mouse IgG was added to nuclear extracts. Samples were incubated overnight at 4°C on a rotating wheel. 30 μl of Protein G Sepharose beads (Cat. no. 17061801; GE Healthcare), previously blocked with 0.5 mg/ml chicken egg albumin, were added to nuclear extracts, followed by a 2-h incubation at 4°C on a rotating wheel. The samples were washed five times in a lysis buffer (20 mM Hepes, pH 7.6, 350 mM KCl, 0.2 mM EDTA, pH 8, 1.5 mM MgCl$_2$, and 20% glycerol) freshly supplemented with 0.5% NP-40 and 0.5 mM DTT. Between each wash, the samples were centrifuged (400$g$, 1 min, 4°C). After the final wash, beads were boiled in Laemmli buffer to solubilise the immunoprecipitated material.

Protein samples were loaded into 10% Bis-Tris Gels (Cat. no. NW00102BOX; Novex) with 1X MOPS SDS running buffer (Cat. no. B0001; Novex). 10 μl of SeeBlue Plus2 prestained protein standard (Cat. no. LC5925; Invitrogen) was used to visualise the protein molecular weight. The electrophoresis was performed at 160 V for 1 h. Proteins were transferred (overnight at 4°C) to a nitrocellulose membrane at 150 mA constant current in the presence of a transfer buffer containing 25 mM Tris, 0.21 M glycine, and 10% methanol. The membrane was blocked for 1 h at room temperature with 10% non-fat skimmed milk dissolved in PBS supplemented with 0.1% Tween. Then, the membrane was incubated for 1 h at room temperature with primary antibodies diluted to the working concentration (see table) in 5% nonfat skimmed milk dissolved in PBS supplemented with 0.1% Tween. The membrane was washed three times with PBS supplemented with 0.1% Tween, and incubated for 2 h at room temperature with LI-COR IRDye–conjugated secondary antibodies diluted 1:5,000 in 5% nonfat skimmed milk dissolved in PBS supplemented with 0.1% Tween. The membrane was finally washed three times with PBS supplemented with 0.1% Tween before analysis with the LI-COR Odyssey FC imaging system.

#### Antibodies

| Antibody | Reference | Working dilution (application) |
|---|---|---|
| Flag | Cat. no. F1804; Sigma-Aldrich | 1:500 (immunofluorescence) |
| Flag | Cat. no. F3165; Sigma-Aldrich | 1:5,000 (Western blot) |
| HA | Cat. no. sc-805; Santa Cruz | 1:50 (immunofluorescence) |
| V5 | Cat. no. 14-6796-80; eBioscience | 1:250 (immunofluorescence), 1:1,000 (Western-blot) |
| NANOG | Cat. no. 14-5761-80; eBioscience | 1:500 (immunofluorescence) |
| NANOG | Cat. no. A300-397A; Bethyl Laboratories | 1:2,000 (Western-blot) |
| Oct6 | Cat. no. sc-11661; Santa Cruz | 1:200 (immunofluorescence) |
| EGFP | Cat. no. Ab13970; Abcam | 1:200 (immunofluorescence) |

### RT–qPCR analysis

Total RNA was isolated using the RNeasy Plus Mini Kit (cat. 74136; QIAGEN), following the manufacturer's instructions. The quantity and purity of RNA samples were determined using a microvolume spectrophotometer (ND-1000; NanoDrop). RNA was reverse transcribed with SuperScript III (Cat. no. 18080044; Invitrogen) using random hexamer oligonucleotides, following the manufacturer's instructions. Triplicate qPCR reactions were set up with the Takyon SYBR MasterMix (Cat. no. UF-NSMT-B0701; Eurogentec) and analysed using the Roche LightCycler 480 machine. For all qPCR primer pairs, standard curves were performed to assess the amplification efficiency and melting curves were generated to verify the production of single DNA species.

### List of qPCR primers

| Primer pairs | Forward primer | Reverse primer |
|---|---|---|
| Esrrb | CGATTCATGAAATGCCTCAA | CCTCCTCGAACTCGGTCA |
| NANOG | AGGATGAAGTGCAAGCGGTG | TGCTGAGCCCTTCTGAATCAG |
| Oct4 | GTTGGAGAAGGTGGAACCAA | CTCCTTCTGCAGGGCTTTC |
| TBP | GGGGAGCTGTGATGTGAAGT | CCAGGAAATAATTCTGGCTCA |
| Tet1 | TTGAGAAGATAGTGTTCACGG | CACTTCTTCTGATCACCCAC |
| Tet2 | CTCATGGAAGAAAGGTATGGAG | GCTCTTGCCTTCTTTACCAG |
| Tet3 | ACTGTCAAGACAGGCTCAG | ATCTCCATGGTACACTGGC |

### ChIP–seq analysis

ChIP-seq datasets were analysed using the Galaxy platform: https://usegalaxy.org/ (40). The bioinformatics workflow is available at the following address: https://usegalaxy.org/u/raf4579/w/workflow-chip-seq-1. Raw sequencing data (FASTQ files) were downloaded from the NCBI's Gene Expression Omnibus database. Quality control was performed using the software "FastQC" (41). Samples were filtered to remove contaminating adapter sequences and low-quality reads (cutoff quality score >20.0). Reads were mapped to the mouse mm9 reference genome using "Bowtie2" (BAM file output) (42). Reads were mapped only to a unique genomic location (k = 1). ChIP-seq peaks were called using the software "MACS2" (BED file output) (43). The immunoprecipitated sample was compared with the genomic input for identifying statistically significant binding sites (q value 0.05). To visualise ChIP-seq datasets on a genome browser, mapped reads (BAM files) were converted into bigwig files using "deepTools" (44). Data were normalised in "reads per kilobase million" (RPKM) to allow the comparison between ChIP-seq datasets. Genomic snapshots were taken using the genome viewer "IGV" (45). To visualise ChIP-seq datasets as heat maps, the software "deepTools" was used (44). RPKM-normalised bigwig files were aligned to ESC enhancers (24).

## Data Availability

Previously published NANOG (24) and TET2 (23) high-throughput sequencing data were obtained from the NCBI's Gene Expression

Omnibus database: NANOG ChIP-seq (GSE44286), TET2 ChIP-seq (GSE57700).

## Supplementary Information

## Acknowledgements

We thank Bertrand Vernay for microscopy support. We thank Claire Cryer and Fiona Rossi for assistance with flow cytometry. We are grateful to Kristian Helin (University of Copenhagen) for sharing Tet2 expression plasmids. This work was funded by a UK Medical Research Council grant MR/L018497/1 to I Chambers. R Pantier was supported by a UK Medical Research Council PhD Fellowship. T Tatar was supported by a Darwin Trust of Edinburgh PhD Fellowship.

### Author Contributions

R Pantier: conceptualization, data curation, formal analysis, validation, investigation, visualization, methodology, and writing—original draft, review, and editing.
T Tatar: investigation and methodology.
D Colby: investigation.
I Chambers: conceptualization, resources, supervision, funding acquisition, investigation, project administration, and writing—original draft, review, and editing.

### Conflict of Interest Statement

The authors declare that they have no conflict of interest.

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
