## [Reviewer comments · Life Science Alliance]

Life Science Alliance

Endogenous epitope-tagging of Tet1, Tet2 and Tet3 identifies TET2 as a naïve pluripotency marker

Raphaël Pantier, Tülin Tatar, Douglas Colby, and Ian Chambers

DOI: <https://doi.org/10.26508/lsa.201900516>

Corresponding author(s): Ian Chambers, Edinburgh, University of

Review Timeline:	Submission Date:	2019-08-07
	Editorial Decision:	2019-08-07
	Revision Received:	2019-08-26
	Editorial Decision:	2019-09-09
	Revision Received:	2019-09-19
	Accepted:	2019-09-19

Scientific Editor: Andrea Leibfried

Transaction Report:

Please note that the manuscript was previously reviewed at another journal and the reports were taken into account in the decision-making process at Life Science Alliance.

Referee #1 Review Received: 8th Jul 19

Report for Author:

In this report, Chambers and colleagues addressed an important question about TETs and resolved their role in naive vs primed states. The evidence presented are very strong and support the general conclusion. Furthermore, this will allow further studies to be carried out by others to establish DNA demethylation in naive pluripotent cells.

Referee #2 Review

Received: 11th Jul 19

Report for Author:

In this manuscript, the authors reported the expression patterns of Tet enzymes (Tet1/2/3) in mouse embryonic stem (ES) cells. Because of the lack of good antibodies for immunostaining, they applied the knock-in strategy to add the different epitope tags to the endogenous Tet genes with the Crispr/Cas9 system. As the result, they established the ES cell line in which all Tet enzymes are labeled with epitope tags. Using this cell line and the control lines, they found that Tet2 expression is heterogeneous and lost after the conversion to the primed state. They revealed that the Tet2 expression is largely merged to the Nanog expression and they interact directly. From these findings, they hypothesized that Tet2 and Nanog may cooperate to regulate ES-specific enhancers.

The function of Tet enzymes in ES cells has been well analyzed. Dawlaty et al reported that Tet2 KO ES cells are fully pluripotent since they efficiently contribute to chimeric embryos after blastocyst injection, suggesting its dispensable role (Dev Cell, 2013). Therefore, the hypothetical function of the cooperation between Tet2 and Nanog might not be important to maintain pluripotency. For the interaction between Nanog and Tet enzymes, Costa et al demonstrated that Tet1 directly interact with Nanog and Tet2 has overlapping function with Tet1 for synergistic action with Nanog in reprogramming of somatic cells to pluripotent state although the direct interaction was not reported for Nanog and Tet2 (Nature, 2013), weakening the novelty of the finding shown in this manuscript.

The weakness of this manuscript is the lack of the functional data that makes it quite descriptive. It is little bit below for the publication in EMBO Rep.

1. For the expression of tet3, the authors introduced HA-tag and found that its expression is undetectable. However, it could be due to the sensitivity of HA-tag/anti-HA Ab. Although they served a control, its expression could be very high. It will be ideal if the authors insert V5-tag to Tet3 and compare the expression levels of Tet1/2/3 in parallel with the same tag/Ab system.
2. To verify the hypothesis of the functional interaction between Nanog and Tet2, the authors re-analyzed the published data of ChIP-seq. Do the Tet2 binding depend on Nanog in these co-occupied sites? It could be assessed by ChIP-seq of Tet2 in Nanog-null ES cells.
3. The proportion of the merge between Nanog and Tet2 should be quantified by FACS or other method.
4. What is the significance of the kinetics of Tet expressions during the transition of ES cells to the primed state? The transition happen in few days in normal development whereas it takes much longer in culture dish.

Referee #3 Review

Received: 1st Aug 19

Report for Author:

In this manuscript Pantier et al, investigated the expression of the TET1, TET2 and TET3 in mouse ESCs. They do this by tagging the endogenous Tet genes using CRISPR/Cas9 in ESCs. Then they evaluated at the single cell level the expression of the TET proteins using FACS and IF, finding that TET1 is expressed in the naïve and primed state, whereas TET2 is mostly expressed in the naïve state. They finally showed that TET2 interacts with Nanog.

The ESCs generated could be of interest for the stem cell field as a tool and probably the epigenetic field as well, as the authors showed in several assays the utility of them.

The experiments are mostly well performed (please see some of the comments). However, I do not find that the manuscript provides any strong biological insight into pluripotency biology. The expression of the TET family in ESC (naïve and primed) has been described before, where KD, ChiP and RNA has been well described. Several manuscripts have shown the RNA levels of the TET family (the authors do not find differences between the expression of protein and RNA). Indeed, the authors used ChIP-seq from a previously published paper where TET2 was detected (on this note: using previously published data is great, but remarks that is not novel the presence of TET2 in ESCs).

Thus, I believe that as a manuscript describing a new tool is a good manuscript (please two comments below to improve the manuscript) however it lacks biological insight.

I believe that the following suggestions may improve the characterisation of the cells that the authors have generated.

- 1) The pluripotency of the targeted cells should be tested.
- 2) I would recommend that mass spec is performed in the tagged ESCs to validate the presence of the tagged TETs. This could be done in the sorted + and - populations.

August 7, 2019

Re: Life Science Alliance manuscript #LSA-2019-00516-T

Prof. Ian Chambers
Edinburgh, University of
MRC Centre for Regenerative Medicine
AFRC Centre for Genome Research University of Edinburgh King's Building, West Mains Road
University of Edinburgh
Edinburgh, SCOTLAND EH9 3JQ

Dear Dr. Chambers,

Thank you for transferring your manuscript entitled "Identification of TET2 as a naïve pluripotency marker by comprehensive epitope-tagging of Tet1, Tet2 and Tet3 alleles" to Life Science Alliance. The manuscript was assessed by expert reviewers at another journal before, and the editors transferred those reports to us with your permission.

The reviewers who assessed your manuscript elsewhere before appreciated the high quality of the work, but would have expected more functional insight. This concern does not preclude publication in Life Science Alliance, and we would thus like to invite you to submit a revised version of your manuscript for publication here. We would expect a point-by-point response to all concerns raised. You already provided an outline of how you would respond to the issues raised upfront, and we think that your responses as well as the proposed quantification and self-renewing assay addresses the reviewer concerns sufficiently. So please proceed as planned.

The typical timeframe for revisions is three months.

Thank you for this interesting contribution to Life Science Alliance. We are looking forward to receiving your revised manuscript.

Sincerely,

Andrea Leibfried, PhD
Executive Editor
Life Science Alliance
Meyerhofstr. 1
69117 Heidelberg, Germany

t +49 6221 8891 502
e a.leibfried@life-science-alliance.org
www.life-science-alliance.org

B. MANUSCRIPT ORGANIZATION AND FORMATTING:

We are pleased that the reviewers generally appreciate our tagging strategy to visualise TET proteins and think that our cell lines would be useful reagents for the scientific community. We note that all reviewers are happy with the quality of our data. We respond to specific reviewer's points below:

Referee #1:

In this report, Chambers and colleagues addressed an important question about TETs and resolved their role in naive vs primed states. The evidence presented are very strong and support the general conclusion. Furthermore, this will allow further studies to be carried out by others to establish DNA demethylation in naive pluripotent cells.

We are pleased that this reviewer appreciates the value of our efforts.

Referee #2:

In this manuscript, the authors reported the expression patterns of Tet enzymes (Tet1/2/3) in mouse embryonic stem (ES) cells. Because of the lack of good antibodies for immunostaining, they applied the knock-in strategy to add the different epitope tags to the endogenous Tet genes with the Crispr/Cas9 system. As the result, they established the ES cell line in which all Tet enzymes are labeled with epitope tags. Using this cell line and the control lines, they found that Tet2 expression is heterogeneous and lost after the conversion to the primed state. They revealed that the Tet2 expression is largely merged to the Nanog expression and they interact directly. From these findings, they hypothesized that Tet2 and Nanog may cooperate to regulate ES-specific enhancers.

The function of Tet enzymes in ES cells has been well analyzed. Dawlaty et al reported that Tet2 KO ES cells are fully pluripotent since they efficiently contribute to chimeric embryos after blastocyst injection, suggesting its dispensable role (Dev Cell, 2013).

TET2 is dispensable for pluripotency/self-renewal (Dawlaty et al., 2013 and our unpublished data). This is why we refer to TET2 in our manuscript title as a "marker" rather than a master regulator of self-renewing ESCs.

Therefore, the hypothetical function of the cooperation between Tet2 and Nanog might not be important to maintain pluripotency. For the interaction between Nanog and Tet enzymes, Costa et al demonstrated that Tet1 directly interact with Nanog and Tet2 has overlapping function with Tet1 for synergistic action with Nanog in reprogramming of somatic cells to pluripotent state although the direct interaction was not reported for Nanog and Tet2 (Nature, 2013), weakening the novelty of the finding shown in this manuscript.

The weakness of this manuscript is the lack of the functional data that makes it quite descriptive. It is little bit below for the publication in EMBO Rep.

Our main aim was to generate ESC lines carrying tagged Tet1/2/3 alleles and perform detailed characterisation of TET protein expression in pluripotent cells as this had not been done properly in past studies, due to the lack of reliable antibodies for TET1/2/3. We discovered that TET2 protein expression is heterogeneous in ESCs cultured in serum/LIF and is tightly associated with the self-renewing capacity of ESCs. This novel functional data intersects well with our recent studies on the functional significance of ESRRB heterogeneity (Festuccia et al., EMBO J 2018).

1. For the expression of tet3, the authors introduced HA-tag and found that its expression is undetectable. However, it could be due to the sensitivity of HA-tag/anti-HA Ab. Although they served a control, its expression could be very high. It will be ideal if the authors insert V5-tag to Tet3 and compare the expression levels of Tet1/2/3 in parallel with the same tag/Ab system.

Using our tagged ESC line, we find that TET3 expression is undetectable by immunofluorescence (Figure 1, panel B). We used a commercial antibody which presents a high affinity for the HA epitope-tag: Santa Cruz Y-11, cat. sc-805. Tet3 transcripts are very low with levels 50-times lower than Tet1 and 10-times lower than Tet2, respectively, in ESCs cultured in serum/LIF (Supplementary Figure 6, panels A and B). Our results with HA-Tet3 are therefore consistent with mRNA studies and our conclusions are unlikely to be altered by performing the V5-Tet3 tagging analysis.

2. To verify the hypothesis of the functional interaction between Nanog and Tet2, the authors re-analyzed the published data of ChIP-seq. Do the Tet2 binding depend on Nanog in these co-occupied sites? It could be assessed by ChIP-seq of Tet2 in Nanog-null ES cells.

A recent study reported that TET2 could be efficiently ChIPed only after extensive crosslinking with “formaldehyde + DSG” (Rasmussen et al, 2019), suggesting that TET2 interacts indirectly with chromatin. TET2 localises to completely different loci in ESCs and hematopoietic cells (Rasmussen et al, 2019), supporting a model in which TET2 is targeted to chromatin via protein-protein interactions in a context-dependent manner. This is consistent with the fact that TET2 does not have a DNA binding domain. Therefore, one might expect transcription factors, such as NANOG to target TET2 to pluripotency enhancers. We amended the Discussion of our manuscript (p.12, second paragraph) to better reflect this point. SOX2 has also been reported to physically interact with TET2 (Zhu et al, 2014). As there are many studies showing an overlap in chromatin binding sites for Nanog and Sox2 (Chen, et al. 2008; Kim et al. 2008 are the cornerstone papers), it is possible that TET2 can be targeted to chromatin also by Sox2, and may do so even in Nanog-null cells. Therefore, it is not straightforward to think that deletion of Nanog would give the clean outcome the reviewer predicts. So, while we agree that the reviewer’s suggestion is indeed an interesting one, we consider this to be beyond the scope of the present study.

3. The proportion of the merge between Nanog and Tet2 should be quantified by FACS or other method.

In our study, we report that TET2 is heterogeneously expressed in ESCs cultured in serum/LIF and correlates with NANOG expression by co-immunofluorescence (Figure 3, panel A). An additional line of evidence in favour of a positive correlation between NANOG and TET2 is that FACS-sorted TET2-positive ESCs are significantly enriched for Nanog transcripts compared to TET2-negative ESCs (Figure 2, panel D). As the reviewer requests, we have now quantitated the number of cells that have negative, single- and double-positive NANOG/TET2 nuclei. This data is presented as a new panel in Figure 3 (panel B) and shows, in line with our prior conclusions, that the majority of cells that express TET2 also express NANOG.

4. What is the significance of the kinetics of Tet expressions during the transition of ES cells

to the primed state? The transition happens in few days in normal development whereas it takes much longer in culture dish.

We used two different protocols to model the transition out of naïve pluripotency: EpiSC (Guo et al, 2009) and EpiLC (Hayashi et al, 2011). Stable EpiSC lines are obtained after multiple passages (Figure 4 and Supplementary Figure 12) whereas ESCs homogeneously differentiate into EpiLC after only 48h (Figure 4 and Supplementary Figure 13). The reviewer states that “transition happens in few days in normal development whereas it takes much longer in culture”. However, this is only true for EpiSCs as the transition times in vivo and for EpiLCs are similar. Importantly, TET2 expression was consistently lost in both EpiLCs and EpiSCs (Figure 4, panels B and C).

Referee #3:

In this manuscript Pantier et al, investigated the expression of the TET1, TET2 and TET3 in mouse ESCs. They do this by tagging the endogenous Tet genes using CRISPR/Cas9 in ESCs. Then they evaluated at the single cell level the expression of the TET proteins using FACS and IF, finding that TET1 is expressed in the naïve and primed state, whereas TET2 is mostly expressed in the naïve state. They finally showed that TET2 interacts with Nanog. The ESCs generated could be of interest for the stem cell field as a tool and probably the epigenetic field as well, as the authors showed in several assays the utility of them. The experiments are mostly well performed (please see some of the comments). However, I do not find that the manuscript provides any strong biological insight into pluripotency biology. The expression of the TET family in ESC (naïve and primed) has been described before, where KD, ChIP and RNA has been well described. Several manuscripts have shown the RNA levels of the TET family (the authors do not find differences between the expression of protein and RNA). Indeed, the authors used ChIP-seq from a previously published paper where TET2 was detected (on this note: using previously published data is great, but remarks that is not novel the presence of TET2 in ESCs).

Thus, I believe that as a manuscript describing a new tool is a good manuscript (please two comments below to improve the manuscript) however it lacks biological insight.

*To our knowledge, most of the data regarding the expression of TET family genes in pluripotent cells (or other cell types) correspond to RNA levels. Even though it may not be surprising that RNA and protein expression correlate quite well, this is a novel discovery. Moreover, assessing the expression of TET proteins in single cells by immunofluorescence and FACS allowed us to reveal the heterogeneity of TET2 expression in ESCs. **This novel finding allowed us to perform subsequent experiments which linked TET2 to self-renewal and naïve pluripotency.***

We also believe that our tagged ESC lines will constitute useful reagents for the scientific community as they will allow further biochemical characterisation of TET family proteins using high-affinity epitope tag antibodies (for ChIP, IP-mass spec, etc.).

I believe that the following suggestions may improve the characterisation of the cells that the authors have generated.

- 1) The pluripotency of the targeted cells should be tested.

All our targeted ESC lines are derived from pluripotent E14Tg2a ESCs (see Supplementary Figure 1). The two main cell lines used in this study are “Tet^{tag/tag}” (Figures 1 and 4) and

“Tet2^{Flag-IRES-EGFP}” (Figures 2 and 3) ESCs. “Tet2^{Flag-IRES-EGFP}” ESCs retain expression of the pluripotency factor NANOG (Figure 3A). We have now added similar analyses showing that “Tet1^{V5/V5}” and “Tet1^{tag/tag}” ESCs also retain expression of NANOG (Supplementary Figure 8B,C). “Tet2^{Flag-IRES-EGFP}” ESCs have a similar self-renewing capacity as wild-type ESCs (Figure 2C). We have now performed a similar self-renewal assay in “Tet1^{V5/V5}” and “Tet1^{tag/tag}” ESCs (Supplementary Figure 8A). This shows that both lines retain a proportionally similar self-renewal capacity to E14Tg2a ESCs.

2) I would recommend that mass spec is performed in the tagged ESCs to validate the presence of the tagged TETs. This could be done in the sorted + and - populations.

In all our ESC lines, correct targeting was verified by PCR genotyping. In particular, we also checked that the TET coding sequence was in-frame with the added epitope tag by Sanger sequencing of the Tet alleles (and we can provide these, if necessary). We therefore consider MS (while interesting) is unnecessary as a “validation” procedure.

September 9, 2019

RE: Life Science Alliance Manuscript #LSA-2019-00516-TR

Prof. Ian Chambers
Edinburgh, University of
MRC Centre for Regenerative Medicine
5 Little France Dr
University of Edinburgh
Edinburgh, SCOTLAND EH16 4UU
United Kingdom

Dear Dr. Chambers,

Thank you for submitting your revised manuscript entitled "Identification of TET2 as a naïve pluripotency marker by epitope-tagging of Tet1, Tet2 and Tet3 alleles". I appreciate your responses and the changes introduced in revision and I would thus be happy to publish your paper in Life Science Alliance pending final revisions necessary to meet our formatting guidelines:

- please add a callout to FigS8 panel C in the text
- please add a scale bar for Fig1D
- please note that some of the white spacers between panels are barely visible/not visible, please check and increase the space slightly

A. FINAL FILES:

- An editable version of the final text (.DOC or .DOCX) is needed for copyediting (no PDFs).
- High-resolution figure, supplementary figure and video files uploaded as individual files: See our detailed guidelines for preparing your production-ready images, <http://www.life-science-alliance.org/authors>
- Summary blurb (enter in submission system): A short text summarizing in a single sentence the

study (max. 200 characters including spaces). This text is used in conjunction with the titles of papers, hence should be informative and complementary to the title. It should describe the context and significance of the findings for a general readership; it should be written in the present tense and refer to the work in the third person. Author names should not be mentioned.

B. MANUSCRIPT ORGANIZATION AND FORMATTING:

Sincerely,

We are pleased that the reviewers generally appreciate our tagging strategy to visualise TET proteins and think that our cell lines would be useful reagents for the scientific community. We note that all reviewers are happy with the quality of our data. We respond to specific reviewer's points below:

Referee #1:

In this report, Chambers and colleagues addressed an important question about TETs and resolved their role in naive vs primed states. The evidence presented are very strong and support the general conclusion. Furthermore, this will allow further studies to be carried out by others to establish DNA demethylation in naive pluripotent cells.

We are pleased that this reviewer appreciates the value of our efforts.

Referee #2:

In this manuscript, the authors reported the expression patterns of Tet enzymes (Tet1/2/3) in mouse embryonic stem (ES) cells. Because of the lack of good antibodies for immunostaining, they applied the knock-in strategy to add the different epitope tags to the endogenous Tet genes with the Crispr/Cas9 system. As the result, they established the ES cell line in which all Tet enzymes are labeled with epitope tags. Using this cell line and the control lines, they found that Tet2 expression is heterogeneous and lost after the conversion to the primed state. They revealed that the Tet2 expression is largely merged to the Nanog expression and they interact directly. From these findings, they hypothesized that Tet2 and Nanog may cooperate to regulate ES-specific enhancers.

The function of Tet enzymes in ES cells has been well analyzed. Dawlaty et al reported that Tet2 KO ES cells are fully pluripotent since they efficiently contribute to chimeric embryos after blastocyst injection, suggesting its dispensable role (Dev Cell, 2013).

TET2 is dispensable for pluripotency/self-renewal (Dawlaty et al., 2013 and our unpublished data). This is why we refer to TET2 in our manuscript title as a "marker" rather than a master regulator of self-renewing ESCs.

Therefore, the hypothetical function of the cooperation between Tet2 and Nanog might not be important to maintain pluripotency. For the interaction between Nanog and Tet enzymes, Costa et al demonstrated that Tet1 directly interact with Nanog and Tet2 has overlapping function with Tet1 for synergistic action with Nanog in reprogramming of somatic cells to pluripotent state although the direct interaction was not reported for Nanog and Tet2 (Nature, 2013), weakening the novelty of the finding shown in this manuscript.

The weakness of this manuscript is the lack of the functional data that makes it quite descriptive. It is little bit below for the publication in EMBO Rep.

Our main aim was to generate ESC lines carrying tagged Tet1/2/3 alleles and perform detailed characterisation of TET protein expression in pluripotent cells as this had not been done properly in past studies, due to the lack of reliable antibodies for TET1/2/3. We discovered that TET2 protein expression is heterogeneous in ESCs cultured in serum/LIF and is tightly associated with the self-renewing capacity of ESCs. This novel functional data intersects well with our recent studies on the functional significance of ESRRB heterogeneity (Festuccia et al., EMBO J 2018).

1. For the expression of tet3, the authors introduced HA-tag and found that its expression is undetectable. However, it could be due to the sensitivity of HA-tag/anti-HA Ab. Although they served a control, its expression could be very high. It will be ideal if the authors insert V5-tag to Tet3 and compare the expression levels of Tet1/2/3 in parallel with the same tag/Ab system.

Using our tagged ESC line, we find that TET3 expression is undetectable by immunofluorescence (Figure 1, panel B). We used a commercial antibody which presents a high affinity for the HA epitope-tag: Santa Cruz Y-11, cat. sc-805. Tet3 transcripts are very low with levels 50-times lower than Tet1 and 10-times lower than Tet2, respectively, in ESCs cultured in serum/LIF (Supplementary Figure 6, panels A and B). Our results with HA-Tet3 are therefore consistent with mRNA studies and our conclusions are unlikely to be altered by performing the V5-Tet3 tagging analysis.

2. To verify the hypothesis of the functional interaction between Nanog and Tet2, the authors re-analyzed the published data of ChIP-seq. Do the Tet2 binding depend on Nanog in these co-occupied sites? It could be assessed by ChIP-seq of Tet2 in Nanog-null ES cells.

A recent study reported that TET2 could be efficiently ChIPed only after extensive crosslinking with "formaldehyde + DSG" (Rasmussen et al, 2019), suggesting that TET2 interacts indirectly with chromatin. TET2 localises to completely different loci in ESCs and hematopoietic cells (Rasmussen et al, 2019), supporting a model in which TET2 is targeted to chromatin via protein-protein interactions in a context-dependent manner. This is consistent with the fact that TET2 does not have a DNA binding domain. Therefore, one might expect transcription factors, such as NANOG to target TET2 to pluripotency enhancers. We amended the Discussion of our manuscript (p.12, second paragraph) to better reflect this point. SOX2 has also been reported to physically interact with TET2 (Zhu et al, 2014). As there are many studies showing an overlap in chromatin binding sites for Nanog and Sox2 (Chen, et al. 2008; Kim et al. 2008 are the cornerstone papers), it is possible that TET2 can be targeted to chromatin also by Sox2, and may do so even in Nanog-null cells. Therefore, it is not straightforward to think that deletion of Nanog would give the clean outcome the reviewer predicts. So, while we agree that the reviewer's suggestion is indeed an interesting one, we consider this to be beyond the scope of the present study.

3. The proportion of the merge between Nanog and Tet2 should be quantified by FACS or other method.

In our study, we report that TET2 is heterogeneously expressed in ESCs cultured in serum/LIF and correlates with NANOG expression by co-immunofluorescence (Figure 3, panel A). An additional line of evidence in favour of a positive correlation between NANOG and TET2 is that FACS-sorted TET2-positive ESCs are significantly enriched for Nanog transcripts compared to TET2-negative ESCs (Figure 2, panel D). As the reviewer requests, we have now quantitated the number of cells that have negative, single- and double-positive NANOG/TET2 nuclei. This data is presented as a new panel in Figure 3 (panel B) and shows, in line with our prior conclusions, that the majority of cells that express TET2 also express NANOG.

4. What is the significance of the kinetics of Tet expressions during the transition of ES cells

to the primed state? The transition happen in few days in normal development whereas it takes much longer in culture dish.

We used two different protocols to model the transition out of naïve pluripotency: EpiSC (Guo et al, 2009) and EpiLC (Hayashi et al, 2011). Stable EpiSC lines are obtained after multiple passages (Figure 4 and Supplementary Figure 12) whereas ESCs homogenously differentiate into EpiLC after only 48h (Figure 4 and Supplementary Figure 13). The reviewer states that “transition happen in few days in normal development whereas it takes much longer in culture”. However, this is only true for EpiSCs as the transition times in vivo and for EpiLCs are similar. Importantly, TET2 expression was consistently lost in both EpiLCs and EpiSCs (Figure 4, panels B and C).

Referee #3:

In this manuscript Pantier et al, investigated the expression of the TET1, TET2 and TET3 in mouse ESCs. They do this by tagging the endogenous Tet genes using CRISPR/Cas9 in ESCs. Then they evaluated at the single cell level the expression of the TET proteins using FACS and IF, finding that TET1 is expressed in the naïve and primed state, whereas TET2 is mostly expressed in the naïve state. They finally showed that TET2 interacts with Nanog. The ESCs generated could be of interest for the stem cell field as a tool and probably the epigenetic field as well, as the authors showed in several assays the utility of them. The experiments are mostly well performed (please see some of the comments). However, I do not find that the manuscript provides any strong biological insight into pluripotency biology. The expression of the TET family in ESC (naïve and primed) has being described before, where KD, ChiP and RNA has been well described. Several manuscripts have shown the RNA levels of the TET family (the authors do not find differences between the expression of protein and RNA). Indeed, the authors used ChIP-seq from a previously published paper where TET2 was detected (on this note: using previously published data is great, but remarks that is not novel the presence of TET2 in ESCs).

Thus, I believe that as a manuscript describing a new tool is a good manuscript (please two comments below to improve the manuscript) however it lacks biological insight.

*To our knowledge, most of the data regarding the expression of TET family genes in pluripotent cells (or other cell types) correspond to RNA levels. Even though it may not be surprising that RNA and protein expression correlate quite well, this is a novel discovery. Moreover, assessing the expression of TET proteins in single cells by immunofluorescence and FACS allowed us to reveal the heterogeneity of TET2 expression in ESCs. **This novel finding allowed us to perform subsequent experiments which linked TET2 to self-renewal and naïve pluripotency.***

We also believe that our tagged ESC lines will constitute useful reagents for the scientific community as they will allow further biochemical characterisation of TET family proteins using high-affinity epitope tag antibodies (for ChIP, IP-mass spec, etc.).

I believe that the following suggestions may improve the characterisation of the cells that the authors have generated.

- 1) The pluripotency of the targeted cells should be tested.

All our targeted ESC lines are derived from pluripotent E14Tg2a ESCs (see Supplementary Figure 1). The two main cell lines used in this study are “Tet^{tag/tag}” (Figures 1 and 4) and

“Tet2^{Flag-IRES-EGFP}” (Figures 2 and 3) ESCs. “Tet2^{Flag-IRES-EGFP}” ESCs retain expression of the pluripotency factor NANOG (Figure 3A). We have now added similar analyses showing that “Tet1^{V5/V5}” and “Tet1^{tag/tag}” ESCs also retain expression of NANOG (Supplementary Figure 8B,C). “Tet2^{Flag-IRES-EGFP}” ESCs have a similar self-renewing capacity as wild-type ESCs (Figure 2C). We have now performed a similar self-renewal assay in “Tet1^{V5/V5}” and “Tet1^{tag/tag}” ESCs (Supplementary Figure 8A). This shows that both lines retain a proportionally similar self-renewal capacity to E14Tg2a ESCs.

2) I would recommend that mass spec is performed in the tagged ESCs to validate the presence of the tagged TETs. This could be done in the sorted + and - populations.

In all our ESC lines, correct targeting was verified by PCR genotyping. In particular, we also checked that the TET coding sequence was in-frame with the added epitope tag by Sanger sequencing of the Tet alleles (and we can provide these, if necessary). We therefore consider MS (while interesting) is unnecessary as a “validation” procedure.

September 19, 2019

RE: Life Science Alliance Manuscript #LSA-2019-00516-TRR

Prof. Ian Chambers
Edinburgh, University of
MRC Centre for Regenerative Medicine
5 Little France Dr
University of Edinburgh
Edinburgh, SCOTLAND EH16 4UU
United Kingdom

Dear Dr. Chambers,

Thank you for submitting your Research Article entitled "Endogenous epitope-tagging of Tet1, Tet2 and Tet3 identifies TET2 as a naïve pluripotency marker". It is a pleasure to let you know that your manuscript is now accepted for publication in Life Science Alliance. Congratulations on this interesting work.

DISTRIBUTION OF MATERIALS:

Again, congratulations on a very nice paper. I hope you found the review process to be constructive and are pleased with how the manuscript was handled editorially. We look forward to future exciting

submissions from your lab.

Sincerely,
